# Arc-like magmas generated by mélange-peridotite interaction in the mantle wedge

E.A. Codillo[1], V. Le Roux [ID] [2] & H.R. Marschall [ID] [3]

The mechanisms of transfer of crustal material from the subducting slab to the overlying mantle wedge are still debated. Mélange rocks, formed by mixing of sediments, oceanic crust, and ultramafics along the slab-mantle interface, are predicted to ascend as diapirs from the slab-top and transfer their compositional signatures to the source region of arc magmas. However, the compositions of melts that result from the interaction of mélanges with a peridotite wedge remain unknown. Here we present experimental evidence that melting of peridotite hybridized by mélanges produces melts that carry the major and trace element abundances observed in natural arc magmas. We propose that differences in nature and relative contributions of mélanges hybridizing the mantle produce a range of primary arc magmas, from tholeiitic to calc-alkaline. Thus, assimilation of mélanges into the wedge may play a key role in transferring subduction signatures from the slab to the source of arc magmas.

[1] Massachusetts Institute of Technology/Woods Hole Oceanographic Institution Joint Program in Oceanography/Applied Ocean Science and Engineering, Woods Hole, Massachusetts 02543, USA. [2] Department of Geology and Geophysics, Woods Hole Oceanographic Institution, 266 Woods Hole Road, Woods Hole Massachusetts 02543, USA. [3] Institut für Geowissenschaften, Goethe Universität Frankfurt, Altenhöferalle 1, 60438 Frankfurt am Main, Germany. Correspondence and requests for materials should be addressed to V.L.R. (email: vleroux@whoi.edu)

Subduction zones are widely studied because they are a major locus of volcanic and seismic hazards. In particular, the compositions of arc magmas have been used to understand the magmatic processes operating in the deep Earth. During subduction, hydrated oceanic crust and sediments are subducted and recycled back into the Earth's interior. Although the fate of subducted sediments is uncertain, their signature is imprinted in the chemistry of most arc magmas around the world[1]. Sediments are globally enriched in many trace elements (e.g., K, Rb, Th, rare earth elements) relative to peridotite mantle[2], thus small volumes of sediments can drastically shift the trace element budget of the mantle wedge. Arc magmas are also characteristically enriched in fluid-mobile large-ion lithophile elements (LILE) such as Ba and Sr, and depleted in high field strength elements (HFSE) such as Nb, relative to mid-ocean ridge basalt (MORB)[3]. The LILE enrichment has usually been attributed to mantle wedge metasomatism by slab-derived fluids[4] produced during dehydration of the subducting slab. The HFSE depleted character, on the other hand, has been attributed to different processes such as a 'pre-subduction' mantle depletion[5,6], selective retention of HFSE by accessory phases (e.g., rutile, sphene, and perovskite) stabilized in the mantle wedge and/or in the slab[7,8], and preferred partitioning of HFSE into orthopyroxene during melt-rock reaction[9]. Although extensive geochemical studies have suggested that arc magma chemistry reflects variable contributions from a depleted MORB mantle (DMM), altered oceanic crust (AOC) and sediments[10,11], experimental studies have faced challenges to simultaneously reproduce both the major and trace element characteristics of tholeiites and calc-alkaline melts, the most common types of arc magmas. In addition, the processes by which typical trace element signatures are produced and transferred to arc magmas remain a matter of debate. In particular, it has been recently argued that the trace element and isotope variability of global arc magmas could not be reconciled with the classic model of arc magma genesis, which invokes hybridization of the mantle wedge by discrete pulses of melted sediments and aqueous fluids from dehydrating AOC. Instead, the trace element and isotope data of global arcs can only be reconciled if physical mixing of sediments + fluids + mantle takes place early on in the subduction process before any melting occurs[12]. This prerequisite redefines the order of events in subduction zones and supports an important role for mélange rocks in arc magmatism. However, the trace and major element chemistry of melts that would result from the interaction of natural mélange rocks with a peridotitic mantle in subduction zones has never been investigated experimentally and remains unknown. Such information is critical to determine whether mélange rocks are viable contributors to arc magmatism worldwide.

Mélange rocks are observed in field studies worldwide[13] and are believed to form by deformation-assisted mechanical mixing, metasomatic interactions and diffusion at different P-T conditions along the slab-mantle interface during subduction[13–16]. Mélanges are hybrid rocks composed of cm to km-sized blocks of altered oceanic crust, metasediments, and serpentinized peridotite embedded in mafic to ultramafic matrices[14,17,18]. These matrix rocks include near-monomineralic chlorite schists, talc schists, and jadeitites with variable amounts of Ca-amphibole, omphacite, phengite, epidote, and accessory minerals (e.g., titanite, rutile, zircon, apatite, monazite, and sulfides), among others. Although the volumes of mélange rocks at depth are poorly constrained, several km-thick low-seismic velocity regions observed at the slab-top in subduction zones worldwide indicate the persistence of hydrated rocks – mélange zones – at the slab-mantle interface[14,15,19]. This km-scale estimate of mélange rocks from

seismic observations is corroborated by numerous field studies of exhumed high-pressure terranes reporting thicknesses ranging from several hundreds of meters up to several kilometers[14,17,20–22]. Mélange rocks display significant spatial heterogeneity, but detailed field observations indicate that chemical potential gradients between juxtaposed lithologies (e.g., metasediments, eclogite, and serpentinized peridotites) may be reduced to homogenous matrices through diffusion and fluid advection processes as mélanges mature[14,23]. For the purpose of this first study, we will assume that mélange matrices are broadly representative of the bulk composition of the mélange and provide a relevant first-order approximation of mélange compositional variability as they form at the expense of, and reflect chemical contributions from their protoliths[15,21]. Although more compositions will be studied in the future, the mélange matrix samples used here reflect two contrasting members in the range of mélange materials that we use to explore possible melt compositions produced by mélange-peridotite interaction.

Laboratory[24] and numerical simulations of subduction process[25–28] have shown that hydration and partial melting may induce gravitational instabilities at the slab-mantle interface, which can develop into diapiric structures composed of partially molten materials. Although these diapirs have not been unambiguously imaged in active subduction zones, we note that along-arc geophysical studies are rare, that the current resolution of seismic techniques may not be appropriate to image mixed mélange-peridotite lithologies, and that magnetotelluric approach, sensitive to interconnected free fluids, would not easily detect the presence of mélanges, where most of the water may be crystallographically bounded. With their intrinsic buoyancy, mélange diapirs have been predicted to form at the slab-top, migrate to the overlying mantle[25,26], and transfer the compositional signatures of slab-derived rocks to the source region of arc magmas[23,29]. In particular, physical mixing and homogenization of viscous mélange diapirs and sub-solidus mantle peridotites is predicted in the hot zones of the mantle wedge[30]. Recent findings on ophiolitic zircon grains also support the idea that material can be transported in the wedge via cold plumes[31]. However, as stated previously, the major and trace element compositions of melts that would be produced by melting of a mélange-hybridized mantle wedge remains unexplored.

Here we present the first experimental study on the generation of arc-like magmas by melting of mélange-hybridized mantle sources. We perform piston-cylinder experiments at 1.5 GPa and 1150–1350 °C and simulate a scenario where mélange materials rise as a bulk[26,32] into the hot corner of the wedge and homogenize with the peridotite mantle (Fig. 1). Using powder mixtures of DMM-like natural peridotite (LZ-1, Supplementary Fig. 1; 85–95 vol. %) and natural mélange rocks from a high-pressure terrane (SY400B, SY325; 5–15 vol. %), we show that experimentally produced glasses display the major and trace element characteristics typical of arcs magmas (e.g., high Ba contents, high Sr/Y ratios, and negative Nb anomaly). Our study provides evidence that the compositional signatures of sediments and fluids, initially imparted to mélange rocks during their formation at the slab-mantle interface, can be delivered to the source region of arc magmas by mixing of mélange materials with mantle wedge peridotites, and variably enhanced during melting of mélange-hybridized peridotite source. We show that depending on the types and relative contributions of mélange materials that hybridize the mantle wedge, the compositions of the melts vary from tholeiitic to calc-alkaline. We further discuss how lithological heterogeneities observed in supra-subduction ophiolites and arc xenoliths could represent direct evidence for peridotite-mélange interactions.

## Results

**Experimental techniques.** We performed piston-cylinder experiments to investigate the composition of melts produced by partial melting of a natural DMM-like peridotite hybridized by small proportions of natural mélange matrix. We used two starting mixes that consisted of homogenized 'peridotite + sediment-dominated mélange matrix' (PER-SED mix) and homogenized 'peridotite + serpentinite-dominated mélange matrix' (PER-SERP mix). Both mélange matrices are fine-grained chlorite schists from Syros (Greece) with estimated water contents between 2–3 wt. %. These two types of natural mélange matrices span a range of compositions that reflect the first-order variability of global mélange rocks in terms of mineralogy (Supplementary Data 1), immobile element chemistry (Fig. 2), and trace element chemistry (Supplementary Fig. 6). As mélange rocks should be volumetrically small compared to peridotite in the mantle wedge, we only added limited volumes (5–15%) of natural mélange matrix to a natural lherzolite powder (85–95%). We note that mélange rocks would not necessarily represent 5–15 vol.% of the sub-arc region at all times because of the 3-D nature of mélange diapirs. Certain regions of the wedge could be hybridized by different amount of mélange materials at different times. Although this experimental design is more challenging because it produces small melt pools, it simulates a more realistic scenario. Experimental melts were collected using glassy carbon spheres placed at the top of Au-Pd capsules. The natural peridotite (LZ-1; from Lherz, France) displays modal proportions and major and trace element compositions similar to DMM (Supplementary Fig. 1). The PER-SED and PER-SERP starting materials were partially melted at 1.5 GPa and 1280–1350 °C, conditions applicable to arc magmatism[33,34]. In addition, near-solidus (1230 °C) and solidus (1150 °C) experiments were performed to better constrain the solid phase assemblage at the beginning of and before melting, respectively. The quenched, dendrite-free glasses were analyzed for major elements using electron microprobe (EPMA) at the Massachusetts Institute of Technology. In addition, chemical maps for major elements were acquired on all experiments (Fig. 3 and Supplementary Fig. 2). Trace element compositions of glass pools were analyzed using a Cameca 3 F secondary ion mass spectrometer (SIMS) at the North East National Ion Microprobe Facility (Woods Hole Oceanographic Institution). Backscattered electron (BSE) images and energy

dispersive spectroscopy (EDS) maps were acquired on all experiments using a Hitachi tabletop SEM-EDS TM-3000. The major and trace element compositions of starting mixes and experimental melts are summarized in Supplementary Data 1 and 2, respectively. We assessed approach to equilibrium by performing a time-series of experiments at 1.5 GPa and 1280 °C, with run durations ranging from 3 h to 96 h. The capsules were preconditioned to minimize Fe loss, although we still observed a decrease in $FeO_T$ (total iron) with increasing run duration. We observed that melt compositions performed between 72 and 96 h were indistinguishable in terms of $SiO_2$, $Al_2O_3$, MgO, $Na_2O$, CaO, $K_2O$, MnO, and $TiO_2$, within 1 s.d. value (Supplementary Fig. 4). Thus, a 72-h run duration was chosen to closely approach equilibrium in those experiments. Mass balance calculations yielded a sum of squared residuals <0.39 (FeO excluded), attesting for a close system for all other major oxides. Phase proportions for each experiment were calculated from mass balance calculations and are reported in Supplementary Data 3. Additional information is provided in the Supplementary Information.

**Phase assemblages.** Solidus (PER-SED 95–5 at 1150 °C) and near-solidus (PER-SED 85–15 at 1230 °C and PER-SERP 85–15 at 1230 °C) experiments were performed at 1.5 GPa to determine the phase assemblages at and near the onset of melting. The solidus experiment produced an assemblage of olivine (ol) + orthopyroxene (opx) + clinopyroxene (cpx) + spinel (sp) ( ± isolated melt drops), while both near- solidus experiments produced silicate melt + ol + opx + cpx + minor sp which typically occurred as small inclusions (~2 μm diameter) in olivine. The compositions of these melts were not analyzed due to their limited exposure. Also, abundant dendrites have compromised their composition. These near- and solidus experiments show that although mélanges have complex mineralogies (Supplementary Data 3), the hybrid peridotite assemblage is standard. At 1280–1315 °C, PER-SED experiments produced the following assemblages: silicate melt + ol + opx + cpx (Supplementary Data 3). Melt proportions vary from 10 to 31%. PER-SERP experiments produced an assemblage of silicate melt + ol + opx + cpx from 1280 to 1350 °C. Melt proportions vary from 3 to 25%. In all experiments, no accessory phase was observed in the chemical maps (Fig. 3 and Supplementary Fig. 2) and high-resolution BSE images (Supplementary Fig. 3). In all 72-hour experiments, an opx-rich reaction band developed along the interface of the accumulated melts and the residue (Fig. 3). The thickness of the opx-rich reaction zone is similar in all experiments, ranging from 125–370 μm. The opx-reaction zone is not present in the 3-hour experiment due to the short run duration (Supplementary Fig. 2). Mineral compositions are homogeneous in individual experiment and show variability between experiments (Supplementary Data 5). Modal proportions in each experiment were calculated using the mass balances (excluding FeO) and are reported in Supplementary Data 3. Representative EDS maps and BSE images are presented in Supplementary Fig. 3.

**Major element composition of the melts.** The major element compositions of hybrid peridotite-mélange melts are reported in Supplementary Data 2 (volatile-free basis). They range in composition from trachyandesite to basaltic trachyandesite and basaltic compositions (49.9–56.5 wt. % $SiO_2$, 2.2–8.8 wt.% $Na_2O$ + $K_2O$) with increasing melting temperatures. The water contents of the melts range from ~ 0.6 to ~ 5 wt.% and were estimated from the difference between major element totals and 100%. The melts also show a large range in $Al_2O_3$ contents (14.8–19.5 wt.%) and MgO contents (7.0–15.9 wt.%) (Supplementary Fig. 5). The melt composition is homogeneous throughout the capsule and

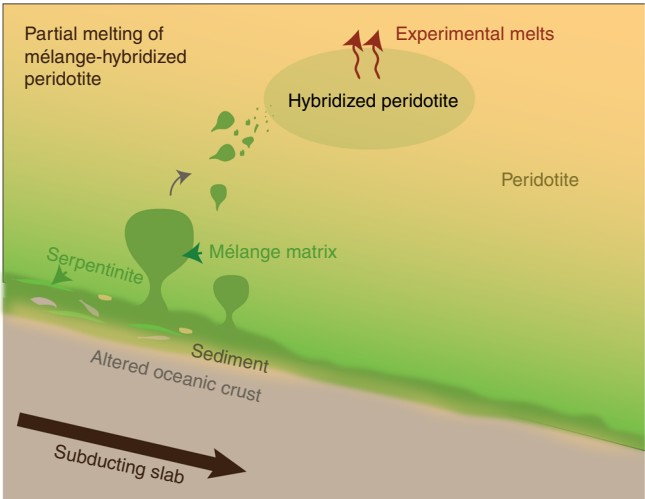

**Fig. 1** Conceptual illustration of mélange-peridotite interaction in subduction zones. Experiments performed in this study simulate the partial melting of a wedge peridotite, hybridized by mélange material that ascended from the slab-top

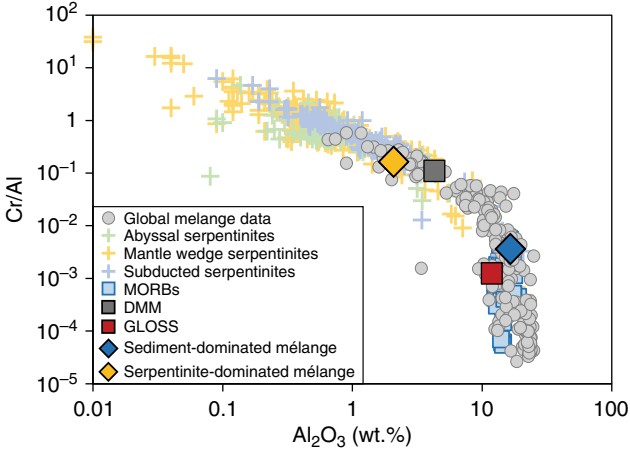

**Fig. 2** Global variability of mélange rocks plotted in terms of immobile elements, Cr/Al versus $Al_2O_3$. The two types of mélange matrices used in this study cover a large range of the natural variability of mélanges, with one matrix that carries a serpentinite/ultramafic flavor, and one matrix that carries a sediment (GLOSS) flavor. Literature data sources: Global mélange compilation[23], abyssal, mantle wedge and subducted serpentinite compilation[81], MORB compilation[58], DMM composition[66], and GLOSS[2]

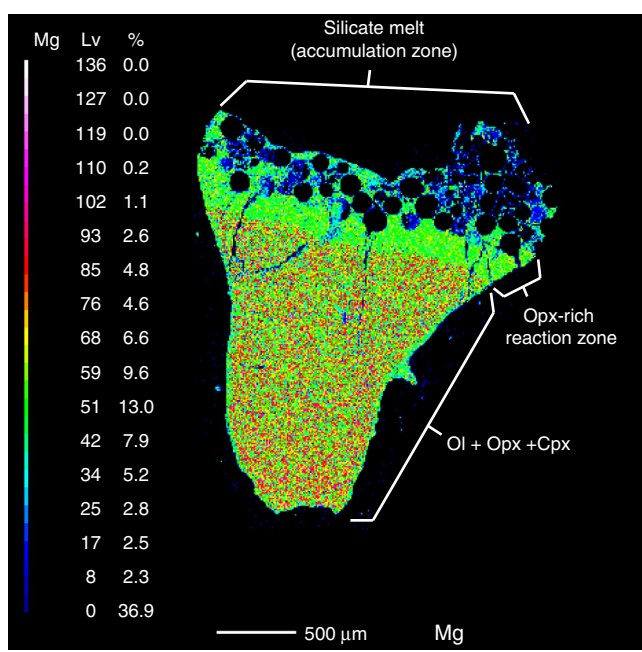

**Fig. 3** Mg compositional map of a representative experiment, PER-SERP (85–15), performed at 1.5 GPa and 1315 °C. An opx-rich reaction zone was observed along the interface between the melt pool and the residual minerals (green band). PER-SERP experiments produced hydrous melt + ol + opx + cpx at all temperatures investigated

does not vary with distance to the Opx-rich reaction zone. The melt compositions from PER-SED and PER-SERP experiments show an increase in CaO, MgO and MnO, a decrease in $K_2O$, $Na_2O$, $Al_2O_3$, $SiO_2$, $TiO_2$, and $P_2O_5$, with increasing temperature. The alkali contents of melts produced from PER-SED experiments are higher than those from PER-SERP experiments at similar temperatures, due to the higher alkali contents of PER-SED starting material (Supplementary Data 1 and Supplementary Fig. 6). The $FeO_T$ contents of peridotite-mélange melts are lower than global arc data due to some limited Fe loss to the capsule

(Supplementary Fig. 7). We now compare the major element compositions of experimental melts (Fig. 4 and Supplementary Fig. 7) with fractionation-corrected global arc data[35] (normalized to MgO = 6 wt.%), primitive arc melts compilations[33,34], and previous experimental studies (Supplementary Data 4). Experimental hydrous peridotite melt compositions[36] match well the major element compositions of global arcs, although alkali contents are expectedly lower than in most arc magmas. Experimental melts from mantle hybridized by slab melts[37] are lower in CaO, and higher in $TiO_2$, $Na_2O$, $K_2O$, and $SiO_2$ compared to arc datasets. Experimental melts from olivine hybridized by sediment melt[38] are lower in CaO, and higher in $Na_2O$ and $K_2O$ compared to arc datasets. Experimental mélange-type 1 and type 2 melts[27,39] are both lower in CaO and MnO and higher in $K_2O$ and $SiO_2$ compared to arc datasets. Interestingly, the major element compositions of experimental mélange-type 2 melts[40], which are partial melts from the sediment-dominated mélange material used in this study (SY400B), plot in the continuity of PER-SED experiments but with higher elemental abundances. Experimental melts from mantle hybridized by sediment melts[41] are higher in $K_2O$ compared to arc datasets. Conversely, partial melts of peridotite hybridized by mélange materials produced in this study plot within or near the compositional field defined by arc datasets for $SiO_2$, MgO, $Na_2O$, $K_2O$, $TiO_2$, $P_2O_5$, and CaO. In terms of alkali contents, lower degree melts (10–19%) of PER-SED experiments are slightly higher than global arcs but plot within the field of global arcs at higher degree of melting (25–31%). Overall, partial melts of peridotite hybridized by mélange materials are similar to partial melts of hydrous peridotites and match well the alkali and major element compositions of typical arcs magmas.

Experimental melts from PER-SED experiments range from the boundary between tholeiitic and calc-alkaline fields to high-K calc-alkaline field (Fig. 5). On the other hand, experimental melts from PER-SERP experiments plot tightly within the tholeiitic field. Experimental mélange-type 2 melts[40] show a strong enrichment in $K_2O$ and plot in the ultrapotassic shoshonitic field. Our results, along with the experimental data of Cruz-Uribe et al.[40], highlight a continuum in alkali enrichment from tholeiitic melts produced by melting of mantle hybridized by serpentinite-dominated mélange, to calc-alkaline melts produced by melting of mantle hybridized by sediment-dominated mélange materials, to ultrapotassic shoshonitic melts from melting of pure sediment-dominated mélange materials.

**Trace element composition of the melts**. The trace element compositions of hybrid peridotite-mélange melts are presented in N-MORB-normalized spider diagrams (Fig. 6) along with global arc data[35], with emphasis on the dominant primitive arc magma types[33] (i.e., calc-alkaline and tholeiite), and published experimental studies that provided both major and trace element contents of experimental melts (Supplementary Data 4). Primitive calc-alkaline arc magmas are geochemically characterized by up to two orders of magnitude higher trace element concentrations compared to primitive arc tholeiites. Pure sediment melts[7] and melts from olivine hybridized by sediment melts[38] have higher trace element concentrations than global arc magmas and display elemental fractionations that are different from global arcs (e.g., Ba/Th, Sr/Nd). Other previous studies[37,40,42] display trace element abundances that plot in the highest range for natural arc magmas, but with major element compositions that are missing CaO or reflect ultra-potassic melts (high $K_2O$). Here we show that, compared to N-MORB, partial melts of hybrid peridotite-mélange materials display enrichment in LILE (e.g., Ba, Th, Sr, K), high LREE/HREE (e.g., Ce/Yb), high LILE/HFSE (e.g., high

Th/Nb, Ba/Nb, and K/Ti), and plot tightly within the trace element fractionation range defined by global arc data (Fig. 7). Experimental melts from PER-SED experiments record elevated trace element concentrations and show fractionations that are characteristic of primitive calc-alkaline magmas. Sr/Nd ratios still fall within the range of global arcs (Fig. 7), although within the lower range of values. Experimental melts from PER-SERP experiments display trace element concentrations that are an order of magnitude lower than melts from PER-SED experiments, and show a slight enrichment in Sr relative to Ce and Nd. In PER-SED experiments, Zr-Hf are slightly enriched compared to Sm and Ti, whereas in PER-SERP experiments, Zr-Hf are not fractionated from Sm and Ti. Trace element concentrations in the melts generally decrease with increasing temperature, consistent with dilution at higher melting extents in the absence of accessory phases that would retain trace elements in the residue. Overall, melts produced from melting of a peridotite source hybridized by mélange rocks (this study) carry trace element signatures typical of natural arc magmas. In particular, peridotite hybridized by serpentinite-dominated and sediment-dominated mélanges produced melts that carry the trace element characteristics of arc tholeiites and calc-alkaline magmas, respectively.

## Discussion

Geodynamic models of rising mélange diapirs have predicted an uneven distribution of mélange rocks in the mantle wedge that involves both complete and incomplete mixing of mélange rocks and peridotites[30]. Our experiments investigate a scenario where the peridotite mantle wedge and limited volumes of mélange rocks are fully mixed and form a new hybrid source that partially melts (Fig. 1). As the extent and volumetric significance of mélange rocks at the slab-mantle interface are still debated, a growing number of studies support their ubiquitous occurrence and importance at the slab-mantle interface. Petrologic modeling[43], numerical instability analysis of subduction zones[44,45], and metamorphic P-T-t histories of exhumed high-pressure mélange terranes[46–48] strongly support the possibility of exhumation of high-pressure rocks through diapirism within the mantle wedge. Further experiments will model how the path of mélange materials would be affected by the thermal structure of individual subduction zones but are beyond the scope of the current study.

For the purpose of this study, we consider that the two end-member mélange matrices from Syros (Fig. 2) offer compositions that represent a reasonable first-order approximation of global mélange variability. Our choice of using natural chlorite schist

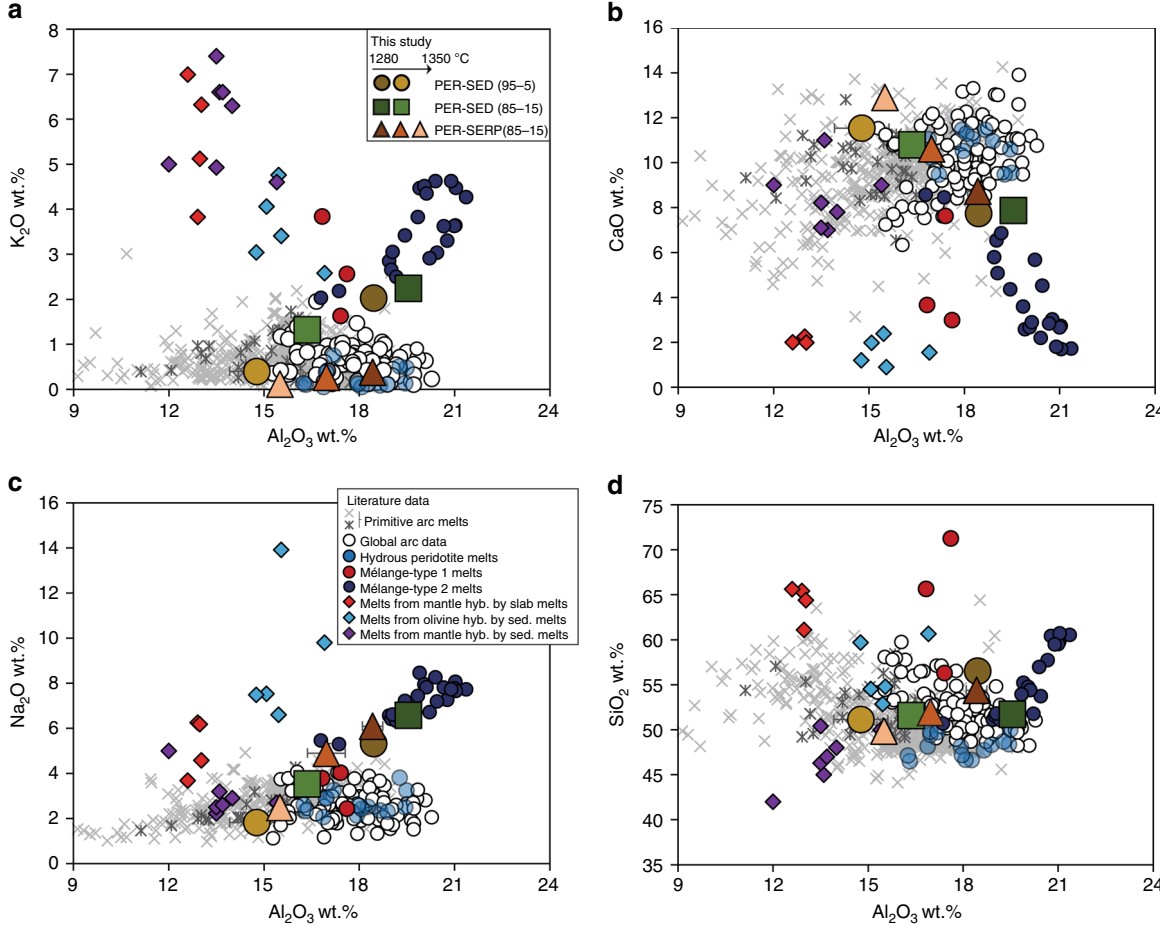

**Fig. 4** Major element composition of experimental melts. Major element variations **a** K$_2$O, **b** CaO, **c** Na$_2$O, **d** SiO$_2$ vs Al$_2$O$_3$ of experimental peridotite-mélange melts from this study compared to global arcs[35] (normalized to MgO = 6 wt.%), two primitive arc melts compilations, and previous experimental studies[37,38,41]. The two primitive arc melts compilations are from Schmidt and Jagoutz[33] (gray asterisk) and Till et al.[34] (light gray cross). Hydrous peridotite melts are from Gaetani and Grove[36]. Experimental melts from mantle hybridized by slab melts and sediment melts are from Rapp et al.[37] and Mallik et al.[41], respectively. Experimental melts of olivine hybridized by sediment melts are from Pirard and Hermann[38]. Experimental mélange-type 1 melts are from Castro and Gerya[27] and Castro et al.[39], while the experimental mélange-type 2 melts are from Cruz-Uribe et al.[40] Our experiments are plotted as averages with error bars representing 1 s.d. All the data, including the literature, are plotted on volatile-free basis

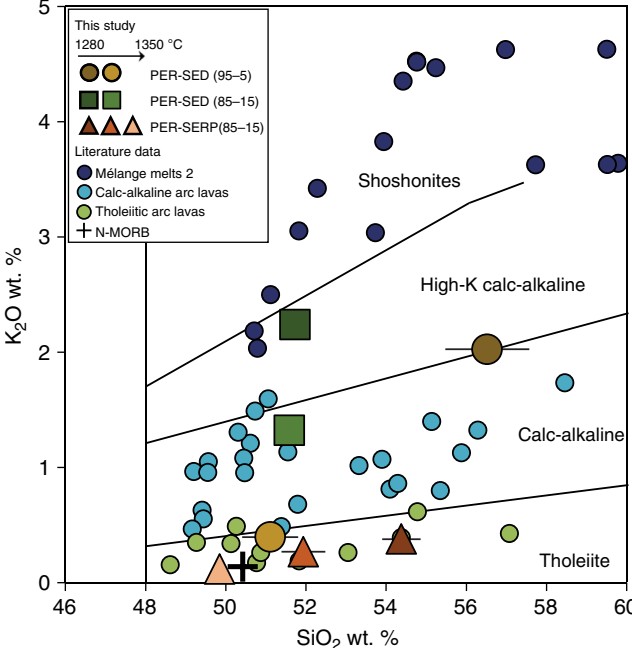

**Fig. 5** $SiO_2$-$K_2O$ diagram highlighting primitive arc magma types and experimental melts from this study. The experimental peridotite-mélange melts from this study are plotted compared to primitive calc-alkaline arc magmas[33] (light blue circles), primitive tholeiitic arc magmas[33] (green circles) and experimental melts of a pure sediment-dominated mélange material (i.e., no interaction with peridotite) of Cruz-Uribe et al.[40] (dark blue circles). Our experiments are plotted as averages with error bars representing 1 s.d. All the data, including the literature, are plotted on volatile-free basis. The average N-MORB value is from Gale et al.[58]

matrix from Syros (Greece) was guided by the fact that the Syros mélange record the mechanical and metasomatic interactions at $P$–$T$ conditions appropriate for slab-mantle interface at depths of about 50–60 km in subduction zones[16,21,23]. In addition, the chlorite ± talc-dominated assemblage in global mélange matrices (including Syros mélange) is relatively insensitive to pressure[49,50], making them a reasonable proxy to the type of mélange extending down to sub-arc depths[14,15,51]. Importantly, our natural starting mélange materials record minimal late-stage modification and overprinting during their exhumation, making their mineralogy, elemental, and volatile concentrations[21] closely approximate the in-situ compositions of mélange rocks at the slab-mantle interface. Thus, the present study offers a reasonable approximation of subduction dynamics where mélange rocks formed at 1.6–2.2 GPa, detach from the slab and homogenizes with peridotite in the hot zones of the mantle wedge at 1.5 GPa (~45 km depth).

Results from our experiments support the idea that primary melts in arcs are not only limited to MgO⁻rich (up to 15.9 wt.%) basalt but may also resemble trachyandesite and basaltic trachyandesite with MgO contents of around 7 wt.% (Supplementary Data 2), covering the MgO range of primitive arc magmas[33]. All of our experiments display CaO, $K_2O$, $Na_2O$, $TiO_2$, and $P_2O_5$ that more accurately reproduce the chemistry of global arc magmas compared to previous studies that simulated hybridization of the wedge by discrete slab melts or discrete sediment melts. The fact that the hybrid source is largely peridotite-like (85–95%) explains why realistic, arc-like major element compositions can be produced in our experiments. Indeed, the large dominance of mantle-equilibrated arc magmas from different subduction zones should reflect the fundamental control of mantle peridotites in controlling the major element compositions of primary arc melts[34,52].

The presence of mall mélange components within the mantle wedge significantly affects the trace element budget of melts generated by melting of a mélange-hybridized mantle source. Although hydrous melting of peridotite would typically produce melts that display a MORB-like trace element pattern[53,54], the trace element compositions of peridotite-mélange melts show striking similarity with global arc magmas, with enriched LILE such as Ba, Th, and K, and depleted HFSE such as Nb and Ti. Previous experimental studies on mantle hybridization by slab melts[37] and sediment melts[42] also produce melts enriched in LILE and depleted in HFSE (Fig. 6d); however their major element compositions mostly reflect (ultra-) potassic shoshonitic melts (high $K_2O$) that occur lesser widely in subduction zones worldwide. Traditionally, melts with high Sr/Y signature have been interpreted as slab melts due to the presence of garnet in the melting residue[55] while the high Th/Nb signature was interpreted to record contribution from sediments melts, as Th can be mobilized more efficiently in sediment melts[56]. In addition, high Ba contents have traditionally been ascribed to addition of fluids[57]. The peridotite-mélange melts plot tightly within the range defined by global arcs for ratios that have traditionally required discrete sedimentary, slab melt, and/or AOC fluid addition to the arc magma source[57]. In particular, the peridotite-mélange melts carry arc-like Sr/Y, Th/Nb, Ba/Th, K/Ti ratios among others (Fig. 7).

In nature, there exists a large compositional variability in primitive arc magmas, ranging from arc tholeiites to calc-alkaline and shoshonites. However, such compositional variability and their spatial distributions (or the lack thereof) have not been satisfactorily explained. Primitive arc tholeiites are usually thought to be produced by decompression style melting (similar to MORB), whereas the classically invoked model for the formation of primitive calc-alkaline magmas envisages their production by flux melting of the mantle induced by the addition hydrous slab components. These slab components are responsible for the up to two orders of magnitude higher trace element concentrations of primitive calc-alkaline magmas relative to N-MORB. For instance, the elevated Th–Zr–$TiO_2$ concentrations of primitive calc-alkaline magmas reflects higher slab contributions in their sources[33]. We have shown that melts produced from melting of a mantle hybridized by sediment-dominated mélanges (PER-SED) strongly resembled primitive calc-alkaline magmas whereas melts produced from melting of a mantle hybridized by serpentinite-dominated mélanges (PER-SERP) strongly resembled primitive arc tholeiites, both in terms of major (e.g., $K_2O$, $TiO_2$) and trace element abundances (e.g., Ba, Th, Zr) and in terms of fractionation characteristics (Fig. 6).

It is critical to determine whether those abundances and fractionations are simply inherited from the starting material or if they are enhanced during melting of the mélange-hybridized peridotite. We make several important observations regarding elemental abundances and fractionations in the melt compared to the starting materials. The bulk starting compositions of PER-SED 95–5 and PER-SERP 85–15 experiments (the two types of experiments that are dominated by ultramafic component – either peridotite or serpentine) fall either outside of the global arc range or within the lower range of values observed in arcs (Fig. 6a, c). It is thus clear that melting plays an important role in producing elemental abundances that are similar to values observed in global arc magmas.

The bulk composition of PER-SED 85–15 experiments (more strongly influenced by a sediment-dominated mélange) is already similar to global arcs for most elements (Fig. 6b), and less surprisingly, melting produces melts that are also similar to arcs. Yet, regardless of abundances, characteristic element ratios acquire a slightly enhanced "arc-like" signature for most elemental ratios

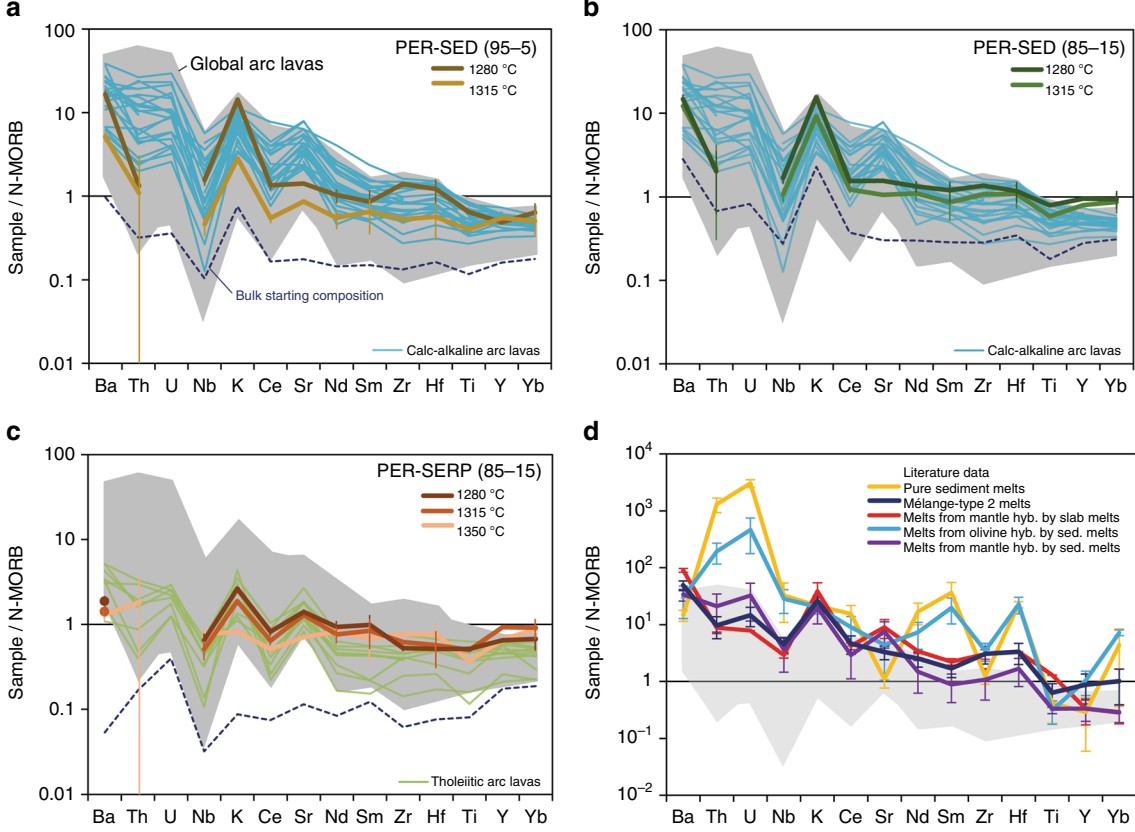

**Fig. 6** Trace element composition of experimental melts Normalized trace element concentrations of experimental peridotite-mélange melts (**a–c**) and previously published experimental studies (**d**), compared to global arcs[35] (normalized to MgO = 6 wt.%; gray field), primitive calc-alkaline[33] (light blue) and arc tholeiites[33] (green). The average N-MORB value used in the normalization is from Gale et al.[58] Data from this study are melt compositions with error bars representing 2SE total (2SE internal error propagated with 2SE from calibration curve). K and Ti were analyzed using EPMA. Bulk starting compositions of the three types of experiments are reported as a dotted blue line. Experimental melts from mantle hybridized by slab melts and sediment melts are from Rapp et al.[37] and Mallik et al.[42], respectively. Experimental melts of olivine hybridized by sediment melts are from Pirard and Hermann[38]. Experimental pure sediment melts are from Skora and Blundy[7]. Experimental melts of a sediment-dominated mélange material (mélange-type 2 melts) are from Cruz-Uribe et al.[40] The literature data are plotted as averages with error bars representing 1 s.d

during melting of mélange-hybridized peridotite. Specifically, Ba/Th, Sr/Y, Zr/Hf, Zr/Sm, and K/Ti slightly increased in melts compared to the starting materials; Ba/Nb, Sr/Nd, and Sm/Nd stay relatively unchanged whereas Th/Nb and Th/Zr slightly decreased compared to the starting materials (Fig. 7).

Experimental melts produced from PER-SED experiments have higher Ba than melts produced from PER-SERP experiments because the sediment-dominated mélange matrix initially had a higher Ba content than the serpentine-dominated mélange matrix (Supplementary Figs. 6 and 8). Still, melts that are produced during melting of PER-SED and PER-SERP starting materials have slightly higher Ba/Th, Sr/Y, Zr/Hf, Zr/Sm, and K/Ti and slightly lower Th/Nb and Th/Zr ratios (compared to starting materials), and thus are not only inherited from the starting materials.

In Supplementary Fig. 9, we show that primitive arc magmas mainly record Nb/Ce$_N$ < 1 (normalized to N-MORB[58]), but their Zr/Sm$_N$ can be below or above 1 and is unrelated to the magma type. In addition to Nb depletion and low Nb/Ce ratios, depletion in Zr and Hf is seen for example in shoshonites from Sulawesi and Fiji, and in calc-alkaline basalts from Solomon and Bismarck[33]. However, Zr, Hf, and Zr/Hf are actually variable in natural primitive arc magmas. Elevated Zr-Hf and Zr/Sm$_N$ ( >1) observed in low-degree melts from PER-SED experiments are features that are observed in natural arc magmas such as calc-alkaline andesites from Japan and New Zealand, calc-alkaline

basalts from Mexico, and depleted andesites from Izu-Bonin. Meanwhile, low-degree melts from PER-SERP experiments have Zr/Sm$_N$ < 1 that overlap with some HFSE-depleted arc magmas such as tholeiitic basalts and andesites from Japan, Cascades and Tonga arcs. We note that in PER-SED experiments, elevated Zr/Sm (and Hf/Sm) does not reflect inheritance from the mélange matrix (Supplementary Fig. 6). Instead, the variability in Zr-Hf contents and Zr/Hf in natural mélange matrices most likely reflect some Zr-Hf mobility in the absence/destabilization of zircon[15]. Overall, the trace element characteristics of our experimental melts plot well within the range of primitive arc magmas (Fig. 7). Thus, these experiments do not only reproduce elemental abundances (major and trace) but also elemental fractionations observed in global arc magmas. In addition, we show that although the trace element compositions of peridotite-mélange melts are partly inherited from the mélanges themselves (i.e., some characteristic subduction signatures may be already imprinted at the slab interface), those arc-like abundances and fractionation signatures can be readily produced and variably enhanced during melting of a mélange-hybridized mantle source (i.e., additional fractionation should occur in the mantle wedge).

Using chemical maps and high-resolution BSE images, we did not observe accessory phases, unlike what had been found in pure mélange melt residues[40]. Our results indicate that elements that have similar incompatibilities during pure peridotite melting can still be slightly fractionated during mélange-hybridized peridotite

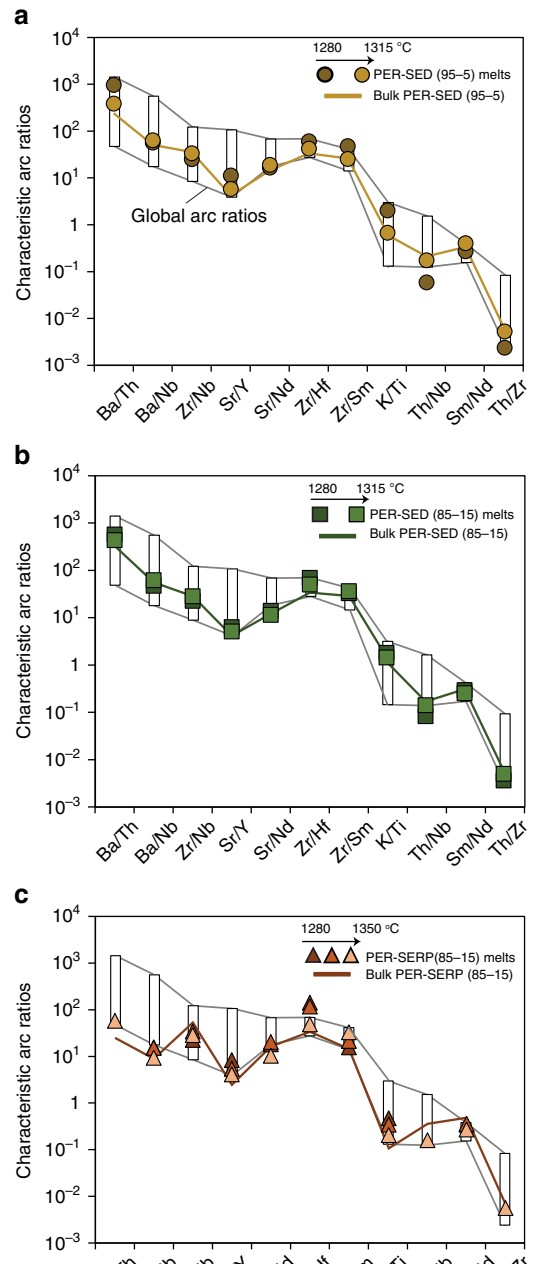

**Fig. 7** Trace element ratios of experimental melts compared to natural arc magmas. Trace element fractionations of experimental peridotite-mélange melts (**a–c**) compared to the bulk starting compositions (yellow, green, and red lines) and global arc ratios defined by Turner and Langmuir database[35] (white rectangles)

melting. Also, we did not observe HFSE- or REE- compatible accessory phases that could retain these elements in the residue. Niobium depletion was in part inherited from the starting bulk compositions (Supplementary Fig. 6) but we hypothesize that it was enhanced by the preferential partitioning of Nb into orthopyroxene[9]. In particular, the presence of an opx-rich reaction zone in all 72-h experiments could have contributed to Nb depletion in the melts. The opx-rich band is likely due to reaction of hydrous melts with the peridotite assemblage, as has been observed in previous studies[59]. Natural pyroxenites, including orthopyroxenites, have been ubiquitously found in exhumed

mantle sections. Previous experimental[27,59] and field-based studies[60–62] have pointed out that orthopyroxenites should form as reaction products of hydrous melts and mantle minerals. The ubiquitous occurrence of orthopyroxenites exposed in suprasubduction zone ophiolites such as in the Josephine[62], Coast range ophiolites[63], and UHP Maowu Ultramafic Complex[64], and sampled in arc-related xenoliths[65], may also potentially record the hybridization of mantle wedge by mélange materials[31]. Thus, the incorporation of mélange diapirs into the mantle wedge may also have implications for the formation of mineralogical and lithological heterogeneities in the mantle.

This study has important implications for the understanding of subduction zone magmatism. During subduction, mélange diapirs may propagate, and dynamically mix with the overlying mantle. Our study shows that depending on the nature and relative contributions of the hybridizing mélange materials in the source of arc magmas, a large variety of primary magmas with characteristic arc-like signatures can be produced. As LILE-enriched shoshonitic melts are expected to form from melting of pure sediment-dominated mélange materials[40], our study shows that both primitive arc tholeiites and primitive calc-alkaline magmas, which are the two most abundant magma types in subduction zones worldwide, can be produced by melting of mantle hybridized by serpentinite-dominated and sediment-dominated mélange materials, respectively. The rarer occurrence of ultrapotassic shoshonites as compared to tholeiites and calc-alkaline magmas likely reflects the volumetric significance of peridotites in the wedge and the dilution effect due to mixing of mélange materials with mantle wedge peridotites. The absence of systematic along- and across-arc spatial distributions of primitive tholeiitic and calc-alkaline arc magmas is consistent with the complexity involved in mélange-diapir ascent paths, and their eventual distributions and mixing with peridotite in the mantle wedge.

In summary, this experimental study provides unique constraints for the role of mélange materials in arc magmatism, as invoked in previous studies. We have shown that melting of a mélange-hybridized peridotite represents a mechanism to generate melts with major element, trace element and trace element ratios characteristic of tholeiitic and calc-alkaline arc magmas. In these experiments, the compositions of starting materials, P–T conditions, and melting degrees were designed to be as realistic as possible compared to natural processes in the mantle wedge. Where mélanges can form and ascend into the wedge, variations in their compositions, thicknesses, and relative contributions in the arc magma source will likely result in the formation of compositionally diverse primary arc melts and can result in the formation of lithological heterogeneities in the mantle. Mélange transfer from the subducting slab to the mantle wedge may be one of several mechanisms by which arc magmas are produced, but we emphasize that both major and trace element of experimental melts need to be reported to better assess how closely we can simulate arc processes. Although further experiments will help decipher the type and amount of mélange materials that could be involved in individual subduction zones, we show that hybridization of peridotite by buoyant mélange rocks is a viable process to transfer crustal signatures from the slab surface to arc magmas.

## Methods

**Starting material preparation.** Alteration-free, natural peridotite (LZ-1; type-locality in Lherz, France) was grinded to a fine powder using agate ball mill. The bulk composition of LZ-1 is similar to DMM[66] in major and trace element compositions (Supplementary Fig. 1) and is here considered to be representative of peridotite mantle wedge. Following grinding, the LZ-1 powder was loaded into a nickel bucket and preconditioned in a 1-atm vertical gas-mixing furnace at 1100 °C with $fO_2$ maintained at the FMQ buffer (Fayalite-Magnetite-Quartz buffer) for 96 h. This $fO_2$ was adjusted by changing the partial pressures of CO and $CO_2$ gases

in the furnace, and is within the range of estimated $fO_2$ for sub-arc mantle[67,68]. Two chlorite schist matrices from Syros (Greece) were selected to represent two end-member compositions of global mélange rocks: the sediment-dominated mélange matrix (SY400B) and the serpentinite-dominated mélange matrix (SY325). Both natural mélange matrices contain water contents of ~2–3 wt. %. We acknowledge that there exists a wide range in chemical and mineralogical compositions of exhumed mélange rocks worldwide and that there is no single rock material that can represent such wide variability. In order to account for this and capture its first-order variability, we selected two mélange matrix rocks from Syros (Greece) based on mineralogical assemblages (Supplementary Data 1), immobile element chemistry (Fig. 1), and trace element chemistry (Supplementary Fig. 5). In Supplementary Data 1, the mineralogical assemblages of SY400B and SY325 are consistent with being derived from a sediment-like and ultramafic/serpentinite-like protoliths, respectively. Using immobile element systematics, Fig. 1 shows a mixing trend between serpentinites and sediment/upper crustal rocks to account for the range in global mélange variability where mélange material SY400B plotted close to GLOSS composition while SY325 plotted close of DMM composition. In Supplementary Fig. 5, the trace element composition of SY400B closely resemble the GLOSS composition while SY325 broadly resemble the DMM-like peridotite, with exception for some highly fluid-mobile elements (e.g., U, K). SY400B and SY325 from Syros record minimal late-stage modification and overprinting during their exhumation, making their mineralogy, elemental and volatile concentrations[21] closely approximate the in-situ compositions of mélange rocks at the slab-mantle interface. Taken together, the mineralogy, immobile element (Cr vs Cr/Al) and trace element chemistry strongly support for the representability of mélange materials SY400B and SY325 to cover for the first-order variability in global mélange composition. Since Syros mélange is one of the most studied and well-constrained exhumed high-pressure mélange terranes in terms of its metamorphic P-T-t condition[69,70] and mélange formation[21,71,72], we have more control on the conditions at which our starting materials have been subjected to and the processes that led to their formation.

These natural mélange materials were grinded to fine powders using agate ball mill. PER-SED 95–5 starting material was prepared using 95 vol.% of LZ-1 and 5 vol.% of SY400B, PER-SED 85–15 was prepared using 85 vol.% of LZ-1 and 15 vol.% of SY400B, and PER-SERP 85–15 was prepared using 85 vol.% of LZ-1 and 15 vol.% of SY325. The peridotite and mélange components were completely homogenized through a thorough process of mixing under ethanol in an agate mortar for 6 cycles, where each cycle is 1 h of grinding. The resulting powder mixes were stored in a dry box until use. Whole-rock compositions (major and trace elements) of LZ-1, SY325, SY400B as well as PER-SED 95–5, 85–15, and PER-SERP 85–15 are summarized in Supplementary Data 1.

**Experimental setup**. Partial melting experiments were performed in 0.5″ end-loaded solid medium piston cylinder device[73] at the Woods Hole Oceanographic Institution (USA). The starting mixes were packed in $Au_{80}Pd_{20}$ capsules conditioned (Fe-saturated) to minimize Fe loss[36]. The $Au_{80}Pd_{20}$ capsules were conditioned by packing MORB-like basalt powder (AII92 29–1) in the capsules and firing them at 1250 °C in a 1-atm vertical gas-mixing furnace with $fO_2$ maintained at 1 log unit below FMQ buffer for 48 h. Ideally, we would have used actual starting materials to condition the capsules, but due to limited quantities of starting materials we considered that a MORB-like basalt would provide enough Fe to saturate the capsules. The silicate glass was removed from the $Au_{80}Pd_{20}$ capsules using warm $HF-HNO_3$ bath.

When loading the starting material into the conditioned $Au_{80}Pd_{20}$ capsules, approximately 35–45 mg of the starting mix was first packed in the capsule and then topped with 3.5–4 mg of vitreous carbon spheres (80–200 μm in diameter) to act as melt entrapments. The capsule was triple-crimped and welded shut. All the experiments were performed in a $CaF_2$ pressure assembly with pre-dried crushable MgO spacers, straight-walled graphite furnace and alumina sleeves. The sealed capsule was strategically positioned in the assembly such that the top portion of the capsule is the side that contains the vitreous carbons spheres to facilitate easy migration of melt during the experiment. Silica powder was placed in between the sealed capsule and alumina sleeve to fill up the space and maintain the capsule's position. Lubricated Pb foils were used to contain the friable $CaF_2$ assembly and to provide lubrication between the assembly and the bore of the pressure vessel.

The experiments were performed at 1280–1350 °C and 1.5 GPa, relevant to arc magma generation[74,75]. Run duration was set at 72 h after verifying approach to equilibrium from a 3 h to 96-h time-series (see paragraph below). Pressure was applied using the cold piston-in technique[76] where the experiments were first raised to the desired pressure before heating them at desired temperature at the rate of 60 °C/min. The friction correction was determined from the Ca-Tschermakite breakdown reaction to the assemblage anorthite, gehlenite, and corundum[77] at 12–14 kbar and 1300 °C and is within the pressure uncertainty ( ± 50 MPa). Thus, no correction was applied on the pressure in this study. Temperature was monitored and controlled in the experiments using $W_{97}Re_3/W_{75}Re_{25}$ thermocouple with no correction for the effect of pressure on thermocouple electromotive force. Temperatures are estimated to be accurate to ±10 °C and pressures to ±500 bars, and the thermal gradient over the capsule was <5 °C. The experiments were quenched by terminating power supply and the run products were recovered. The capsules were longitudinally cut in half before mounting in epoxy. All the mounted

capsules were polished successively on 240- to 1000-grit SiC paper before the final polishing on nylon/velvet microcloth with polycrystalline diamond suspensions (3–0.25 μm) and 0.02 μm colloidal silica. Vacuum re-impregnation of capsules with epoxy was performed to reduce plucking-out of the vitreous spheres during polishing.

**Approach to equilibrium**. Approach to equilibrium was evaluated by performing a time-series of experiments using PER-SED (95–5) starting material at 1.5 GPa and 1280 °C at varying run durations of 3 h, 24 h, 48 h, 72 h, and 96 h. We performed the time-series experiments at the lowest temperature used in this study (1280 °C) such that it provided the maximum amount of time necessary to approach equilibrium. We observed that the melt compositions obtained after 72 h to 96 h were indistinguishable within 1 s.d. values in terms of $SiO_2$, $Al_2O_3$, MgO, $Na_2O$, CaO, $K_2O$, MnO, $TiO_2$, and $P_2O_5$ (Supplementary Fig. 4). It has been shown experimentally that hydrous melting of peridotite produces melts with lower FeO* (~6 wt. %) contents than anhydrous equivalents (~8 wt. %)[36] but we also observed a decrease in $FeO_T$ with increasing run duration, which suggests Fe loss. We speculate that this Fe loss/depletion reflects one or a combination of the following causes: (1) Fe diffusion to the $Au_{80}Pd_{20}$ capsule due to incomplete Fe saturation during conditioning; (2) formation of orthopyroxenite reaction zone, which could have further contributed to Fe depletion. Other observation that indicates a close approach to equilibrium in our experiments is the homogenous distribution of minerals in the matrix away from the reaction zone, and homogeneous major element compositions within single capsule.

Another way of assessing equilibrium between the melt and minerals, and check whether the experiment behaved as a closed system, is based on the quality of mass balance calculations performed for all the major elements. Using the MS-Excel optimization tool Solver, we obtained low values for the sum of squared residuals (<0.39), for all the major elements, except for Fe, attesting for a close approach to equilibrium for all other major oxides in our experiments, and confirming a small amount of Fe loss in the capsule walls. Phase proportions for each experiment estimated from the mass balance calculations were verified visually in every experiment.

**Electron microprobe analysis**. Major element compositions of the quenched melts and coexisting silicate minerals from all experimental run products were analyzed using JEOL JXA-8200 Superprobe electron probe micro-analyzer at Massachusetts Institute of Technology. Analyses were performed using a 15 kV accelerating potential and a 10 nA beam current. The beam diameter varied depending on the target point. For quenched melt pools, beam diameters varied between 3 μm to 10 μm (mostly 5 μm) depending on the size of the melt pools. For silicate minerals, a focused beam (1 μm) was used. Data reduction was done using CITZAF package[78]. Counting times for most elements were 40 s on peak, and 20 s on background. In order to prevent alkali diffusion in glass, Na was analyzed first for 10 s on peak and 5 s on background. All phases (melt and coexisting minerals) were quantified using silicate and oxide standards. The compositional maps for different major elements were performed at similar instrumental setup using a focused beam. Major element compositions of melts and minerals are reported in Supplementary Data 2 and 5, respectively.

**Secondary ion mass spectrometry**. Concentrations of selected trace elements in melt pools (usually <30 μm diameter) were obtained using a Cameca IMS 3f ion microprobe at the Northeast National Ion Microprobe Facility (NENIMF) at the Woods Hole Oceanographic Institution (WHOI). Analyses were done using $^{16}O^-$ primary ion beam with 8.4 keV voltage, 500 pA to 1 nA current and ~10 μm beam diameter. No raster was used in the beam. Positive secondary ions are accelerated to a nominal energy of 4.5 keV. $^{30}Si$ was set as the reference isotope and ATHO-G, T1-G, StHs6/80-G glasses were used as standards[79]. Analyses were carried out in low mass resolution (m/δm = 330) with an energy offset of −85 V. Secondary ions were counted by an electron multiplier. A 1800 μm diameter field aperture size was used for most of the measurements. We did not use the field aperture to block any of the ion image since the spot was already very small. Each measurement consists of four minutes of pre-sputtering, then five cycles with an integration of 10 s/cycle for $^{30}Si$ and 10 s/cycle for elements $^{88}Sr$, $^{89}Y$, $^{90}Zr$, $^{93}Nb$, $^{138}Ba$, and 30 s/cycle for $^{140}Ce$, $^{143}Nd$, $^{147}Sm$, $^{174}Yb$, $^{180}Hf$, $^{232}Th$, and $^{238}U$. Th concentrations are reported if 1SE error is above detection limit. 1SE error for U is below detection limit for all measurements so U is not reported. In total, each analysis spot requires a total analysis time of approximately 60 min. Reduced trace element concentrations of melts obtained by SIMS are reported in Supplementary Data 2. Internal errors from analyses (2 SE) and error from calibration curves (2SE) have been propagated and are incorporated in the total 2SE error reported in the figures and Datasets.

**X-ray fluorescence technique (XRF)**. Whole-rock elemental concentrations of LZ-1, SY400B, and SY325 were analyzed using X-ray fluorescence technique for major elements and inductively coupled mass spectrometer technique for trace elements at GeoAnalytical Laboratory at Washington State University. As stated before, whole-rock compositions (major and trace elements) of LZ-1, SY325,

SY400B as well as PER-SED 95–5, 85–15, and PER-SERP 85–15 are summarized in Supplementary Data 1.

**Major element variability of residual phases**. Major element compositions of residual minerals are homogeneous through the capsule in individual experiments, and vary between experiments due to differences in temperature and starting compositions (Supplementary Fig. 10). They are within the range of values observed in peridotites worldwide, although Fe loss probably artificially increased Mg# of minerals (93–96 in olivine; 91–95 in clinopyroxene; 92–95 in orthopyroxene). Temperature (1280–1350 °C) has variable effect on mineral compositions. For example, with increasing temperature, olivines display a slight decrease in $Al_2O_3$, a constant CaO, and a slight increase in MgO. $D_{MgO}$ ol/melt decreases with increasing temperature. Orthopyroxenes display a slight decrease in $TiO_2$ and $Al_2O_3$ with increasing temperature, whereas $SiO_2$ and MgO increase, and CaO is constant. As predicted experimentally, $D_{Al2O3}$ opx/melt decreases with increasing temperature and $D_{Na2O}$ opx/melt increases with increasing temperature[80]. Clinopyroxenes show limited major element variability between all experiments. The mineral compositions are reported in Supplementary Data 5.

**Data availability**. The authors declare that all data generated during this study are included in this published article (and its Supplementary Information Files).

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

## Acknowledgements

This project was supported by the WHOI Ocean Exploration Institute (OEI) 27071178 to V.L.R; Previous related projects were supported by NSF EAR-1348063 and WHOI OEI to H.R.M; We thank Brian Monteleone and Kathryn Rose Pietro for their help with SIMS analyses. V.L.R. thanks Ed Stolper, Mike Baker, and Glenn Gaetani for helpful discussions. We are also thankful for Emily Sarafian's help during preparation of the experiments.

## Author contributions

V.L.R. and E.A.C. contributed equally. V.L.R. designed the study, collected the peridotite rocks, and provided supervision to E.A.C.; E.A.C. and V.L.R. performed the experiments, E.A.C. and V.L.R. performed the chemical analyses, H.R.M. provided mélange rocks. All authors contributed to the manuscript.

## Additional information

**Competing interests:** The authors declare no competing interests.

