## [Peer Review File · Nature Communications]

Reviewers' comments:

Reviewer #1 (Remarks to the Author):

This manuscript describes experimental results suggesting that arc magmas can be produced in the mantle-wedge, assuming that some mélangé material, rising from the subduction channel, is assimilated with peridotite in hot regions at 1.5 GPa and 1280-1350°C. This process, not really novel, has been investigated since the '80s. (e.g., Sekine & Wyllie, 1982). However, authors claim that the composition of their experimental melts is much more similar to natural rocks compared to previous works. I have the feeling that the authors should put more efforts in order to make their work really novel and thus suitable for Nature Communications. In particular, I am puzzled by the lack of any geochemical and/or petrological model that could have been useful to describe and understand the observed element fractionation, the phase stabilities and the melting process. The manuscript, as it is, appears too descriptive. Further concerns are:

- 1) the choice of natural rocks as starting material, which are prone to metastability and disequilibrium. Can authors envisage some less-complex model systems and use other more effective starting materials (gels, glasses, reagents) in order to better define which are the major players in the described process?
- 2) the representativity of the chosen rock compositions (in particular Syros rocks) on a global scale;
- 2) the approach to equilibrium, as reaction rims between melt pools and peridotite are often observed;
- 3) the stability of sapphirine, which is extremely rare in mantle rocks. I suggest to perform thermodynamic calculations (e.g, pseudosections) in order to constrain the expected P-T-X conditions of formation of this phase.
- 4) the choice of P, T conditions; for instance, the Gerya & Yuen thermal model does not predict >1200°C in the corner flow at subarc conditions.
- 5) some literature experimental melts in the hydrous peridotite and/or metasomatised peridotite systems are not reported and compared with the new experimental results.
- 6) authors say that they cannot investigate accessory minerals because they are only few microns in size. They should try again identifying these phases by means of electron microscopy. They should also provide some BSE images of the observed microtextures, in addition to chemical maps.
- 7) the mineral chemistry of the phases in equilibrium with melts is not discussed properly.
- 8) in the abstract, authors stress the importance of thier study for the trasfer of volatiles, but volatiles are never mentioned in text. It is to underline, however, that both water and carbon (vitreous carbon glass) have been introduced in the starting material.

In conclusion, I cannot recommend immediate publication of this manuscript in Nature Communications. Nevertheless, the experimental results appear promising and the topic is certainly of potential interest for the journal.

Reviewer #2 (Remarks to the Author):

The manuscript "Arc-like magmas generated by melange-peridotite interaction in the mantle wedge" from Codillo et al. aims at understanding the mechanisms of mass transfer from deep subduction zones and the origin of arc magmas formation. The authors present a new, high-quality dataset from an original experimental setup along with extensive analytical results acquired using high resolution analytical tools. This carefully-written manuscript deals with two timely points of debate in the community, namely the origin of arc magmas and the existence of cold diapirs above subduction zones. I have a number of concerns regarding the hypotheses on which these experiments are based on. I also have comments about the interpretations of this interesting

preliminary dataset. After reading the manuscript I believe that these results should be expanded and re-submitted in a longer format where the reader can directly access to the extensive amount of data which is now unfortunately buried in the supplementary material. This expanded version will also give the opportunity to the authors to perform complementary experiments and better discuss the similarities and differences between their results and the abundant literature on the subject. I also believe that some moderation in the writing style would be welcome since the bases on which this work is settled may not be as robust as what the authors claim.

The main caveat of this work is that it critically relies on a thermo-mechanically modeled process (namely the "cold plumes") which has neither been observed in nature nor documented by geophysical means. Similarly the spatial extension of "hybrid rocks" advertised by Marschall and colleagues is absolutely unknown. We ignore whether this mixture is 10m or 5km thick. Continuous field exposures in Syros or in Dominican Republic are too scarce to draw conclusions on the actual volume of these hybrid rocks above the interface. This uncertainty has major implications for the mixing ratios considered here. These cold plumes (if they exist) would surely not represent 5 vol.% of the sub-arc region at a fixed moment of time. The underlying question is: what is the rock volume concerned by this sub-arc melting and what would be the effective melt composition escaping upwards? If a 1 km³ cold plume melts, does it mean that c.20 km³ of the host mantle coevally melts to respect the 95-5 ratio fixed in these experiments? Since we completely ignore how these diapirs physically mix within the host mantle I find a bit hazardous drawing hypotheses based on simple linear mixing of pelitic and ultramafic end-members. Having said that, I am aware that this type of simplification is needed to address such a complex problem. My concern is that the geochemical results presented here may just represent one solution of the problem. Arc patterns shown in figure 6 have been successfully explained by previous experimental works without invoking cold diapirs. My friendly advice to the authors to strengthen their theory would be to search for an experimental strategy to rule out the "classical" model of mantle wedge metasomatism by slab-derived fluids. For instance, the choice of a lherzolite as a starting mantle material is debatable. I believe that a harzburgite would also do the job since in reality there has already been some melt extraction when the alleged "cold plume" comes in and mixes in the sub-arc region. Most mantle wedge xenoliths worldwide are harzburgitic or dunitic in composition. This is just one example to show that there is a broad range of parameters that shall be more systematically explored before drawing strong conclusions like yours on the origin of arc magmas.

As a conclusion, there is some really good work done here. But I think that major improvements are needed to better convince the reader. The short Nature communication format may not be the best option to satisfactorily discuss this complex geological problem.

Minor points:

- Symbol colors have been inverted in figure 2 for blue and yellow symbols.
- L.84: "from A high pressure terrane": I recommend the use of singular here because the authors only use Syros as natural example
- L.141: Opx rims for PER-SED (95-5) did not form because experimental running time was too short compared to other runs! If you want to compare the thickness of the product phase, you should not change more than one parameter (i.e. the temperature).

Original reviewer's comments are copied in black, and our responses are in green. Line numbers in green refer to line numbers in the revised "**Tracked changes**" manuscript.

Reviewers' comments:

Reviewer #1 (Remarks to the Author):

This manuscript describes experimental results suggesting that arc magmas can be produced in the mantle-wedge, assuming that some mélange material, rising from the subduction channel, is assimilated with peridotite in hot regions at 1.5 GPa and 1280-1350°C.

This process, not really novel, has been investigated since the '80s. (e.g., Sekine & Wyllie, 1982).

We appreciate the opportunity to clarify how novel our study is.

No previous experimental study has used natural mélange material to investigate the compositions of melts produced by a mélange-hybridized peridotite wedge. What has been investigated for a long time are discrete additions of fluids, slab melts or sediment melts, that are each responsible for specific signature in arc magmas (eg, high Ba/Th comes fluids, Th/Nb signature traces sediment addition, high Sr/Y could be slab melts etc.). We show here for the first time that mélange diapirs have the potential to produce all these various geochemical signatures at once when added to the mantle wedge. Sekine and Wyllie (1982) (1982), for example, experimentally investigated a scenario where discrete slab melts interact and hybridize the mantle wedge peridotite, stabilizing phlogopite. However, the reactions investigated in that study were done in synthetic systems and did not reproduce the natural data (e.g., from Sekine and Wyllie 1982: "The model liquid differs from the natural magma by the absence of (Ca + Fe + Mg), the absence of Na, and the much higher content of K.").

In addition, by compiling studies that had reported both major and trace element compositions of their experimental melts, we show that no previous experimental study had accurately reproduced the major, trace and trace element fractionation of tholeiitic and calc-alkaline magmas, the two most abundant magmas in arcs.

We have re-emphasized the novel aspects of our study, and how it differs from previous experimental studies in the revised manuscript (l. 56-58; l. 234-243; l. 272-276; l. 358-361; l. 442-456).

However, authors claim that the composition of their experimental melts is much more similar to natural rocks compared to previous works.

We have shown that melting of peridotite hybridized by limited volumes of mélange rocks produces melts that carry the major (Figs. 4, 5 and S7) and trace element abundances (Fig. 6), and fractionated trace element ratios (Fig. 7) characteristic of natural arc lavas. Fig 4 and 6d illustrate how discrete slab fluids, slab melts or sediment melt additions, as simulated by previous experimental studies, could not reproduce some key aspects of the natural data (ie, a number of either trace and/or major elements are off).

I have the feeling that the authors should put more efforts in order to make their work really novel and thus suitable for Nature Communications. In particular, I am puzzled by the lack of

any geochemical and/or petrological model that could have been useful to describe and understand the observed element fractionation, the phase stabilities and the melting process. The manuscript, as it is, appears too descriptive.

We have addressed these concerns as thoroughly as possible, through additional experiments, additional analyses, and additional discussion, as described below.

- *Regarding phase stability.* Although it was not specifically requested, we performed additional experiments in near-solidus (1230°C) and solidus (1150°C) conditions to double-check whether accessory phases (that were not seen as residual phases in our previous experiments) could have controlled the fractionation and trace element budget of the experimental melts. The new experiments are now reported in the text (l. 148-150) and show that the residual assemblage is only controlled by Ol+Opx+Cpx+Spinel (l. 177-185; l. 427-430), confirming our previous results.
- *Regarding trace element fractionation.* While the previous comment helped confirm the solidus and residual assemblages, we re-measured all previous and new experiments for trace elements by SIMS to double-check if any fractionation could be linked to precision/accuracy on these challenging analyses of small melt pools. The trace element contents of melts were previously measured by SIMS Cameca 1280 using a 1-standard (NIST) calibration curve. In the past few months, our facility at WHOI has developed an improved method for measuring trace elements by SIMS using a Cameca 3f ion microprobe instead. Although this was also not specifically requested, we took advantage of these improvements to re-measure all experiments using a 3-standard calibration curve (now reported in Methods, and l. 153-154). The overall improved data quality has allowed us to present Nd data with this revised version. We also now report 2SE (standard error), not 1SE in Fig.6. Note that when one uses a 1-standard calibration curve, the 2SE uncertainty associated with the calibration curve is limited because there is only 1 standard. Here, by using a 3-standard calibration curve, we get more accurate data because we do not rely on one standard only, but our error bars are larger due to the fact that we propagate errors from both the internal 2SE (associated with individual analysis) and a 2SE from the calibration curve. We show that, while the new values do not change our conclusions, the trace element patterns are generally smoother, and trace elements better follow a dilution pattern as degree of melting increases, which supports the lack of accessory phases that would retain some of these elements in the residue. (l. 287-289)
- *Regarding geochemical/petrological model.* In this revised version, we evaluate our results not only in terms of the major and trace element abundances and fractionations compared to global arc lavas as a whole, but also in terms of the compositions and spatial distributions of the different magma types (e.g., tholeiites, calc-alkalines, shoshonites) that occur in all subduction zones worldwide. In particular, we take better advantage of the recent arc compilation by Schmidt and Jagoutz (2017). Our revised manuscript highlights our new observations on the compositions of peridotite-mélange melts. In particular, we show that melts produced from melting of a mantle hybridized by sediment-dominated mélanges (PER-SED experiments) strongly resembled primitive calc-alkaline lavas while melts produced from melting of a mantle hybridized by serpentinite-dominated mélanges (PER-SERP experiments) strongly resembled primitive

arc tholeiites, both in terms of major and trace element abundances, and fractionation characteristics. To this end, we have added sections that significantly strengthen our case for a global application of this model. (e.g., l. 252-260; l. 280-293; l. 372-386; l. 442-456).

Further concerns are:

1) The choice of natural rocks as starting material, which are prone to metastability and disequilibrium. Can authors envisage some less-complex model systems and use other more effective starting materials (gels, glasses, reagents) in order to better define which are the major players in the described process?

Mélange rocks are indeed complex and we made sure to emphasize this in the original manuscript. If one wanted to synthesize a simplistic mélange rock, they would still need to know the natural variability of metastable mélanges from the field, and decide which key chemical characteristics to keep in the synthesized materials. The criticism would be that we do not know which phase/lithology may be controlling the trace element fractionation during melting, and that by simplifying complex rocks, we cannot get the full picture of their roles in arc magmatism. We have expanded why we chose to use mélange matrices as representative compositions for mélange rocks (l.71-91). We believe that, although the chemistry of exhumed mélanges could be in theory slightly different from the ones in-situ in subduction zones, using natural mélanges provides the best analog to determine their roles in arc magmatism.

2) The representability of the chosen rock compositions (in particular Syros rocks) on a global scale;

We have better explained the choice of Syros mélange in order to minimize potential issues described in the previous comment (l. 135-138; l. 323-332).

2) The approach to equilibrium, as reaction rims between melt pools and peridotite are often observed;

In addition to the time-series experiments already presented in the original manuscript, we now report that the melt compositions are not changing depending on the distance of the Opx-reaction zone, confirming approach to equilibrium for these melts (l. 213-214). We also report that Opx compositions are similar in the reaction zone and in the residual assemblage and that mineral compositions are homogeneous throughout the capsule (l. 198-199). We now present the result of mass balance calculations that attest for a close system for all elements (l. 168-169), except some limited FeO loss as previously described in the original manuscript.

3) The stability of sapphirine, which is extremely rare in mantle rocks. I suggest to perform thermodynamic calculations (e.g, pseudosections) in order to constrain the expected P-T-X conditions of formation of this phase.

We are grateful for that comment. We performed SEM again on the 1350°C experiments that contained sapphirine. We did some higher resolution detailed mapping of all the borders of the capsules to double check that Al could not have been introduced by the surrounding alumina sleeve (part of the experimental setup). Although this was checked in the first round, one observation had been overlooked. Through this re-mapping, we noticed in localized spots that

the Al sleeve was in direct contact with the melt in both PER-SED 95-5 and PER-SED 85-15 experiments at 1350°C. Our hypothesis is that very localized melting of capsule (in higher melt fractions areas) allowed for the interaction of the melt and surrounding alumina sleeve, which saturated the melt with Al and enabled the crystallization of sapphirine. That observation had also been undetected from the trace element analyses of the melt as sapphirine plays a limited role, if any, in trace element budget. In the revised version, we have removed PER-SED 95-5 and PER-SED 85-15 experiments performed at 1350°C. The PER-SERP 85-15 1350 °C experiment was confirmed to have an intact capsule as originally thought, so it is still presented in the manuscript (it contains no sapphirine). The smaller amounts of melt in PER-SERP experiments at 1350 °C seem to have helped preservation of the capsule. All capsules from all other experiments were also double-checked at higher resolution using the SEM and confirmed to be intact as originally described.

Since we only had one experiment left for PER-SED 95-5 after we removed the 1350 °C experiment, we performed an additional 72-h PER-SED 95-5 experiment at 1315°C such that we would have both 1280 and 1315 °C for PER-SED 95-5 starting material. This new experiment was also analyzed using the new 3-standard calibration curve technique on the 3f ion microprobe.

4) The choice of P, T conditions; for instance, the Gerya & Yuen thermal model does not predict >1200°C in the corner flow at subarc conditions.

Our experimental *P-T* range for generation of arc magmas (1280-1350 °C) is within the range of global mantle-melt equilibration conditions (~1075–1450 °C at ~0.8–1.9 GPa) calculated by Till (2017) for arc magmas worldwide using a new internally consistent reverse fractionation calculations and thermobarometry for a representative subset of the global primitive arc lavas. It is also within the tighter range of 1.0–2.5 GPa, 1220–1350°C reported by Schmidt and Jagoutz (2017) for tholeiitic and calc-alkaline basalts. Thermal structures provided by numerical models are very sensitive to input parameters, thus we believe that the temperatures of our experiments (1280-1350°C) still fall within a reasonable range of *P-T* conditions for arc magmas. We have emphasized these two studies in the revised manuscript (l. 148).

5) Some literature experimental melts in the hydrous peridotite and/or metasomatised peridotite systems are not reported and compared with the new experimental results.

We agree that there have been numerous experimental studies simulating mantle wedge hybridization by discrete slab-derived components. However, most studies have only reported the major element compositions of their experimental melts. In order for us to be able to compare studies that have attempted to reproduce both the major and the trace element abundances observed in arc magmas, we reported all (to our knowledge) experimental studies that, aside from having simulated mantle wedge hybridization by slab-derived components, have also provided both the major and trace element compositions of their experimental melts.

6) Authors say that they cannot investigate accessory minerals because they are only few microns in size. They should try again identifying these phases by means of electron microscopy. They should also provide some BSE images of the observed microtextures, in addition to chemical maps.

In the process of performing new experiments, we have performed a significant number of additional higher resolution BSE images and EDS mapping using a Hitachi tabletop SEM-

EDS TM-3000, as well as additional eprobe mapping (Jeol 8200; MIT). We took the opportunity to perform additional mapping on our previous experiments too. We have added EDS and BSE examples of typical textures and residual assemblages in the supplementary material (Fig S3; l.). As previously described, we still did not observe any accessory phases in the residual assemblage, therefore we confirm our original observations that no accessory phase has stabilized in these experiments. In addition, the new solidus and near-solidus experiments also confirmed the lack of any accessory phase (l. 177-203) in the starting material, and in the near-solidus residue. Finally, trace elements follow a dilution pattern with increasing degree of melting that does not support a control by accessory phases (l. 288-290).

7) The mineral chemistry of the phases in equilibrium with melts is not discussed properly.

The major element composition of the mineral phases are used in conjunction with the composition of the melt to assess the mass balance for each major element in all the phases in our experiments (l. 169) and are reported in the supplementary material. Mineral compositions are found to be homogeneous throughout the capsule (l. 199). We have added a description of major element variability of minerals in the supplementary material. Thank you for that suggestion.

8) In the abstract, authors stress the importance of their study for the transfer of volatiles, but volatiles are never mentioned in text. It is to underline, however, that both water and carbon (vitreous carbon glass) have been introduced in the starting material.

Agreed. We removed the mention of volatiles as we did not specifically discuss volatiles in the rest of the manuscript. The presence of carbon (glassy carbon spheres) in our experiments may in theory produce hydrous melts with slightly larger amounts of dissolved carbon. However, glassy vitreous carbon spheres have been observed to stay much more intact than, e.g. amorphous carbon spheres, and represent a reasonable approach to trap small fractions of melts. Ideally, we would have used diamonds but that has its own challenges. It is very hard to polish an experiment that contains diamonds evenly, and so it is problematic to expose very small melt pools like in our experiments.

In conclusion, I cannot recommend immediate publication of this manuscript in Nature Communications. Nevertheless, the experimental results appear promising and the topic is certainly of potential interest for the journal.

Thank you for the encouragements and very constructive criticism.

Reviewer #2 (Remarks to the Author):

The manuscript “Arc-like magmas generated by melange-peridotite interaction in the mantle wedge” from Codillo et al. aims at understanding the mechanisms of mass transfer from deep subduction zones and the origin of arc magmas formation. The authors present a new, high-quality dataset from an original experimental setup along with extensive analytical results acquired using high resolution analytical tools. This carefully-written manuscript deals with two timely points of debate in the community, namely the origin of arc magmas and the existence of cold diapirs above subduction zones. I have a number of concerns regarding the hypotheses on which these experiments are based on. I also have comments about the interpretations of this interesting preliminary dataset. After reading the manuscript I believe that these results should be

expanded and re-submitted in a longer format where the reader can directly access to the extensive amount of data which is now unfortunately buried in the supplementary material. This expanded version will also give the opportunity to the authors to perform complementary experiments and better discuss the similarities and differences between their results and the abundant literature on the subject. I also believe that some moderation in the writing style would be welcome since the bases on which this work is settled may not be as robust as what the authors claim.

The main caveat of this work is that it critically relies on a thermo-mechanically modeled process (namely the “cold plumes”) which has neither been observed in nature nor documented by geophysical means.

Mélanges have been observed in the field in numerous places, and numerical models predict their existence at the slab-mantle interface. It is true that mélange diapirs mixed within the wedge have not unambiguously been imaged through geophysics means. However, we note that along-arc geophysical studies are rare, that the current resolution of seismic techniques is probably not appropriate to image mixed peridotite-mélange lithologies, and that magnetotelluric approach, sensitive to interconnected free fluids, would not easily detect the presence of mélanges, where most of the water may be crystallographically bounded (l. 95-99). Some geophysics studies have nonetheless detected a 2-8km low velocity zone right above the slab (Abers 2005), and that zone could correspond to the presence of pure mélange. Finally, we cite a newly published study that argues that ophiolitic zircon have been transported from the slab to the wedge via cold plumes (Proenza et al. 2018).

Similarly the spatial extension of “hybrid rocks” advertised by Marschall and colleagues is absolutely unknown. We ignore whether this mixture is 10m or 5km thick. Continuous field exposures in Syros or in Dominican Republic are too scarce to draw conclusions on the actual volume of these hybrid rocks above the interface. This uncertainty has major implications for the mixing ratios considered here. These cold plumes (if they exist) would surely not represent 5 vol. % of the sub-arc region at a fixed moment of time. The underlying question is: what is the rock volume concerned by this sub-arc melting and what would be the effective melt composition escaping upwards? If a 1 km³ cold plume melts, does it mean that c.20 km³ of the host mantle coevally melts to respect the 95-5 ratio fixed in these experiments? Since we completely ignore how these diapirs physically mix within the host mantle I find a bit hazardous drawing hypotheses based on simple linear mixing of pelitic and ultramafic end-members. Having said that, I am aware that this type of simplification is needed to address such a complex problem.

We agree that we have limited constraints on the actual volumes of mélange materials in the mantle wedge. Our experimental setup investigates a scenario where mélange materials rise as a bulk into the hot corner of the wedge and homogenize with the peridotite mantle. The mélange materials would not necessarily represent 5-15 vol. % of the sub-arc region at all times because of the 3-D nature of mélange diapirs. Certain regions of the wedge could be hybridized by different amount of mélange materials (here we speculate 5-15% to be conservative) at different times. We have added a statement to clarify this aspect in the revised manuscript (l. 140-143).

We believe that questions such as: (1) what is the rock volume concerned by this sub-arc melting and (2) what would be the effective melt composition escaping upwards? are very stimulating questions, but are beyond the scope of the current study. As the reviewer pointed out,

such simplification in our experimental design is necessary to obtain meaningful preliminary results to address a new complex problem.

My concern is that the geochemical results presented here may just represent one solution of the problem.

To our knowledge, no previous experimental study has in fact reproduced simultaneously the major, trace elements, and trace element fractionations of tholeiites and calc-alkaline magmas as closely as what we present here. We have re-emphasized the novel aspects of our study, and how it differs from previous experimental studies in the revised manuscript (l. 56-58; l. 234-243; l. 272-276; l. 358-361; l. 442-456).

We agree that there may not be just one solution as subduction zones are very complex, but here we provide the first mélange-peridotite experiments where we show that at the global scale, mélanges could play a significant role in arc magmatism, as suggested by other independent approaches. We moderated that aspect in the conclusion (l. 466-469).

Arc patterns shown in figure 6 have been successfully explained by previous experimental works without invoking cold diapirs.

As pointed out above, trace elements and/or major elements of previous experimental studies that have used either discrete slab or discrete sediment melts as the hybridizing agents do not simultaneously reproduce both the major and trace element systematics of global arc magmas, and tholeiites and calc-alkaline in particular.

My friendly advice to the authors to strengthen their theory would be to search for an experimental strategy to rule out the “classical” model of mantle wedge metasomatism by slab-derived fluids. For instance, the choice of a lherzolite as a starting mantle material is debatable.

Geochemical evidence in a large meta-data study presented by Nielsen & Marschall (2017) clearly demonstrated that the “classical” model of mantle wedge metasomatism is incompatible with the trace-element and isotope ratios observed in all global arc magmas. The failure of the classical models to explain the generation of arc magmas therefore calls for a change of paradigm, and for the testing of the new mélange diapir models by the means of geophysical, geochemical and indeed experimental petrologic methods. The study presented here is the first step in that new direction. We will definitely be taking this comment by the reviewer as an encouragement to think about ways to discriminate the classical mantle-metasomatism model from the new mélange-diapir model through further petrologic experiments.

I believe that a harzburgite would also do the job since in reality there has already been some melt extraction when the alleged “cold plume” comes in and mixes in the sub-arc region. Most mantle wedge xenoliths worldwide are harzburgitic or dunitic in composition.

We welcome this the reviewer’s suggestion on the use of a more refractory mantle composition as a proxy of the mantle wedge for future experiments. However the use of harzburgites as a proxy for mantle wedge may not drastically affect the trace element compositions and the fractionation characteristics observed in our experimental melts. In Fig. S5, we show that the trace element concentrations of mélange materials are up to two orders of magnitude higher than the natural peridotite. Thus, the trace element budget may mostly be affected by the nature and amounts of mélange materials.

This is just one example to show that there is a broad range of parameters that shall be more systematically explored before drawing strong conclusions like yours on the origin of arc magmas.

As mentioned before, we agree that subduction zones are very complex and that several mechanisms may be at play (l. 465-468).

As a conclusion, there is some really good work done here. But I think that major improvements are needed to better convince the reader. The short Nature communication format may not be the best option to satisfactorily discuss this complex geological problem.

Thank you for the very stimulating comments.

Minor points:

- Symbol colors have been inverted in figure 2 for blue and yellow symbols.

Done.

- L.84: “from A high pressure terrane”: I recommend the use of singular here because the authors only use Syros as natural example

Done.

- L.141: Opx rims for PER-SED (95-5) did not form because experimental running time was too short compared to other runs! If you want to compare the thickness of the product phase, you should not change more than one parameter (i.e. the temperature).

We revised the description. Thicknesses are within the same range in all 72h experiments (ie there is not a clear control of T given the uncertainty in thickness estimates). However, run duration plays a role in the thickness of the reaction zone as 3h experiments have no opx-reaction zones.

REVIEWERS' COMMENTS:

Reviewer #1 (Remarks to the Author):

This revised version of the manuscript satisfactorily addresses most of the criticisms I highlighted in the previous round. In my opinion, the manuscript now meets the standards required to be published in Nature Communications. However, I still have some minor suggestions.

1) line 54-58: I think that the sentence should be de-personalized in order to be consistent with the style of the manuscript and of the journal, highlighting the previous results and not the names of the authors.

2) line 58: AOC is not introduced

3) line 96: the rise of mélange rocks into the mantle wedge has been also suggested by Tumiati et al. (2013) *J Petrol*, who also provide P-T conditions of melting for metasomatized peridotite in the presence of H₂O and CO₂. They also provide major element compositions for near solidus melts, (trachyandesite at low pressures). These experimental data could be compared with author's results.

4) line 152: please replace FeOT with "total iron"

5) line 161: please refer to experiment below solidus as subsolidus (not solidus) experiments.

6) line 195: "This does not affect the conclusions of the study"; please try to make this sentence less axiomatic. In fact, the Fe/Mg ratio is widely considered a key parameter in natural rocks.

7) Discussion: please consider to split this very long paragraph into smaller independent paragraphs.

8) line 281: "MgO-rich basalt (up to 15.9 wt.%)", do you mean "MgO-rich (up to 15.9 wt.%) basalt"?

9) Supplementary Information: because you were using a nickel container during fO₂ conditioning at 1100°C, did you observe Ni contamination/iron loss in the rock powder compared to initial composition? Did you analyse the powders after preconditioning? Moreover, I recommend that all the original data discussed in the manuscript are provided as tables, at least as Supplementary Information. The text parts of the Supplementary Information should be transferred to the main text (a "Methods" section is required at the end of the manuscript), leaving only figures and tables as supplementary material.

Reviewer #2 (Remarks to the Author):

The manuscript from Codillo et al. submitted to Nature communications has been only marginally improved after the correction stage. Even though I am still wondering how relevant and pertinent is the mélange model to explain arc signatures, I have the feeling that the set of data presented by the authors is satisfactorily supporting their model.

This carefully-written contribution might be seen as one step forward to improve our understanding of the deep melting issue. Overall I am not very enthusiastic with this (speculative) model, but I have nothing against acceptance in Nature communications.

Original reviewer's comments are copied in black, and our responses are in green.

Reviewer #1 (Remarks to the Author):

This revised version of the manuscript satisfactorily addresses most of the criticisms I highlighted in the previous round. In my opinion, the manuscript now meets the standards required to be published in Nature Communications. However, I still have some minor suggestions.

1) line 54-58: I think that the sentence should be de-personalized in order to be consistent with the style of the manuscript and of the journal, highlighting the previous results and not the names of the authors. **Done.**

2) line 58: AOC is not introduced. **Done.**

3) line 96: the rise of mélange rocks into the mantle wedge has been also suggested by Tumiati et al. (2013) J Petrol, who also provide P-T conditions of melting for metasomatized peridotite in the presence of H₂O and CO₂. They also provide major element compositions for near solidus melts, (trachyandesite at low pressures). These experimental data could be compared with author's results.

Tumiati et al. (2013) focused on the subsolidus phase equilibria relation of the system K₂O-Na₂O-CaO-FeO-MgO-Al₂O₃-SiO₂ (KNCFMAS) + COH, which is different from investigating the compositions of partial melts from a mélange+peridotite starting material at various degrees of melting. Only one experimental run (ST19) produced an ultrapotassic trachyandesitic melt (K₂O = 9.17 wt. %, SiO₂ = 58.07 wt. %) in equilibrium with ol + opx + grt + cpx but they did not analyze the melt for trace element composition. We chose to compile studies that specifically investigated melt compositions and reported both major and trace elements.

4) line 152: please replace FeOT with "total iron" **Added.**

5) line 161: please refer to experiment below solidus as subsolidus (not solidus) experiments.

The experiments contain local tiny droplets of melts as described in the text and thus cannot be described as subsolidus. Thus, we prefer to describe them as 'solidus'.

6) line 195: "This does not affect the conclusions of the study"; please try to make this sentence less axiomatic. In fact, the Fe/Mg ratio is widely considered a key parameter in natural rocks. **We have removed that sentence.**

7) Discussion: please consider to split this very long paragraph into smaller independent paragraphs. NC does not allow sub-headings but we have broken up the first paragraph of the discussion into two separate paragraphs. The rest of the discussion was already broken up in multiple paragraphs.

8) line 281: "MgO-rich basalt (up to 15.9 wt.%)", do you mean "MgO-rich (up to 15.9 wt.%) basalt"? **This has been modified to clarify. Thank you.**

9) Supplementary Information: because you were using a nickel container during fO₂ conditioning at 1100°C, did you observe Ni contamination/iron loss in the rock powder compared to initial composition? Did you analyse the powders after preconditioning?

The powders were not analyzed post preconditioning. However, the same Ni buckets are regularly used to condition ultramafic material and Fe loss is not linked to that specific step. Also, it is unlikely that at these temperatures, Ni or Fe would have time to diffuse in/out of the dry unmelted powder to the bucket wall.

Moreover, I recommend that all the original data discussed in the manuscript are provided as tables, at least as Supplementary Information.

All data are accessible as excel tables in the supplementary material, such that anyone can easily plot and reproduce our figures.

The text parts of the Supplementary Information should be transferred to the main text (a "Methods" section is required at the end of the manuscript), leaving only figures and tables as supplementary material. **Done.**

Reviewer #2 (Remarks to the Author):

The manuscript from Codillo et al. submitted to Nature communications has been only marginally improved after the correction stage. Even though I am still wondering how relevant and pertinent is the mélange model to explain arc signatures, I have the feeling that the set of data presented by the authors is satisfactorily supporting their model. This carefully-written contribution might be seen as

one step forward to improve our understanding of the deep melting issue. Overall I am not very enthusiastic with this (speculative) model, but I have nothing against acceptance in Nature communications.

Thank you. We hope that our study will invite more discussions in the scientific community to further investigate the importance and potential roles of mélangé rocks not only on mass transfer processes in subduction zones but also on the rheology and deformation at the slab-mantle interface.